
# Mapping the performance of a versatile water-based condensation particle counter (vWCPC) with COMSOL simulation and experimental study

Weixing Hao[1], Fan Mei[2,*], Susanne Hering[3], Steven Spielman[3], Beat Schmid[2], Jason Tomlinson[2], Yang Wang[1,*]

[1]Department of Chemical, Environmental and Materials Engineering,
University of Miami, Miami, FL, 33146, USA

[2]Pacific Northwest National Laboratory, Richland, WA, 99352, USA

[3]Aerosol Dynamics Inc., Berkeley, CA, 94710, USA

*Correspondence to*: Fan Mei (fan.mei@pnnl.gov), Yang Wang (yangwang@miami.edu)



**Abstract.**
Accurate airborne aerosol instrumentation is required to determine the spatial distribution of ambient aerosol particles,
particularly when dealing with the complex vertical profiles and horizontal variations of atmospheric aerosols. A
versatile water-based condensation particle counter (vWCPC) has been developed to provide aerosol concentration
measurements under various environments with the advantage of reducing the health and safety concerns associated
with using butanol or other chemicals as the working fluid. However, the airborne deployment of vWCPCs is relatively
limited due to the lack of characterization of vWCPC performance at reduced pressures. Given the complex
combinations of operating parameters in vWCPCs, modeling studies have advantages in mapping vWCPC
performance.
In this work, we thoroughly investigated the performance of a laminar flow vWCPC using COMSOL Multiphysics®
simulation coupled with MATLAB. We compared it against a modified commercial vWCPC (vWCPC Model 3789,
TSI, Shoreview, MN, USA). Our simulation determined the performance of particle activation and droplet growth in
the vWCPC growth tube, including the supersaturation, $D_{\mathrm{p,kel,0}}$ (smallest size of particle that can be activated),
$D_{\mathrm{p,kel,50}}$ (particle size activated with 50% efficiency) profile, and final growth particle size $D_{\mathrm{d}}$ under wide operating
temperatures, inlet pressures $P$ (0.3 – 1 atm), and growth tube geometry (diameter $D$ and initiator length $L_{\mathrm{ini}}$). The
effect of inlet pressure and conditioner temperature on vWCPC 3789 performance was also examined and compared
with laboratory experiments. The COMSOL simulation result showed that increasing the temperature difference ($\Delta T$)
between conditioner temperature $T_{\mathrm{con}}$ and initiator $T_{\mathrm{ini}}$ will reduce $D_{\mathrm{p,kel,0}}$ and the cut-off size $D_{\mathrm{p,kel,50}}$ of the
vWCPC. In addition, lowering the temperature midpoint ($T_{\mathrm{mid}} = \frac{T_{\mathrm{con}} + T_{\mathrm{ini}}}{2}$) increases the supersaturation and slightly
decreases the $D_{\mathrm{p,kel}}$. The droplet size at the end of the growth tube is not significantly dependent on raising or lowering
the temperature midpoint but significantly decreases at reduced inlet pressure, which indirectly alters the vWCPC
empirical cut-off size. Our study shows that the current simulated growth tube geometry ($D$ = 6.3 mm and $L_{\mathrm{ini}}$ = 30
mm) is an optimized choice for current vWCPC flow and temperature settings. The current simulation can more
realistically represent the $D_{\mathrm{p,kel}}$ for 7 nm vWCPC and also achieved a good agreement with the 2 nm setting. Using
the new simulation approach, we provide an optimized operation setting for the 7 nm setting. This study will guide
further vWCPC performance optimization for applications requiring precise particle detection and atmospheric aerosol
monitoring.



## 1 Introduction

Aerosols, defined as any solid or liquid particles suspended in air, are one of the fundamental components of the atmosphere and have a significant impact on air quality, climate change and human health (Seinfeld et al., 2016; Anderson et al., 2020; Lighty et al., 2000; Pöschl, 2005; Prather et al., 2020; Li et al., 2017; Paasonen et al., 2013; Darquenne, 2012). However, accurate and comprehensive monitoring of aerosol particles is challenging because aerosol particle sizes and number concentrations vary widely both spatially and temporally (Davidson et al., 2005; Yu and Luo, 2009; Krudysz et al., 2009). Airborne measurements and characterization, therefore, are often required to capture the vertical profiles and horizontal variability of atmospheric aerosols.

In understanding the variability of atmospheric aerosol and determining the size distribution and number concentration of aerosols, laminar-flow, butanol-based Condensation Particle Counters (CPCs) used in conjunction with differential mobility analyzers (DMAs) can provide real-time measurements of airborne particles and are widely used by effectively exploiting the working principle of condensation growth (Hermann et al., 2007; Kangasluoma and Attoui, 2019; Mordas et al., 2008; Sem, 2002; Wiedensohlet et al., 1997). However, conventional butanol CPCs face difficulties in characterizing particles below 3 nm in size. In addition, health and safety risks, such as the odor, flammability and toxicity of the butanol, are an issue for many deployments in offices, homes, aircraft, and other inhabited locations. These limitations have led directly to researchers designing advanced aerosol instruments that can be more widely used in both atmospheric environments and laboratory studies.

In 2005, Hering and Stolzenburg (2005) developed a continuous-flow, water-based laminar condensation particle counter (WCPC) (Hering et al., 2005; Hering and Stolzenburg, 2005), which uses distilled water as the working fluid to avoid the health and safety concerns. It was also found to have comparable performance to butanol-based CPCs in previous studies (Biswas et al., 2005; Franklin et al., 2010; Iida et al., 2008; Kupc et al., 2013; Liu et al., 2006; Mordas et al., 2008). A modified version of the WCPC featuring an additional new moderator section has been developed (Hering et al., 2014). With this new moderated approach, the initiator provides water vapor for particle activation while the moderator provides distance and time for particle growth. This improved water CPC achieves the same peak supersaturation and similar droplet growth while reducing the water vapor, particle loss, and side effects of heating the flow in the earlier version of WCPC. Furthermore, a versatile WCPC was then developed capable of particle detection near 1 nm without using a filtered sheath flow. The operating temperatures can also be adjusted in accordance with the cut-point desired (Hering et al., 2017).

Since water-based CPCs have comparable performance to butanol-based CPCs while also offering the advantage of avoiding health and safety risks, it is desirable to explore advanced water-based CPCs in a broader range of environmental and energy applications. To improve the detection performance of the vWCPC, we need to identify the effects of operational factors and geometry. However, the limited analysis of relevant temperature and geometric parameters in the vWCPC makes it challenging to control condensational growth conditions. In addition, the inlet pressure effect is another critical factor affecting the detection efficiency of vWCPCs. The potential of using vWCPC





for airborne deployment or other lower pressure monitoring has not been fully explored. Mei et al. (2021) found that
the counting efficiency of the vWCPC 3789 operated at the factory settings decreased with decreasing the operating
pressure, particularly at operating pressures below 700 hPa. However, determining how to reduce the lower detection
limit under various ambient pressures also needs to be investigated.

Due to the complex matrix of geometry, operating temperature, and inlet pressure parameters in vWCPCs, modeling
studies are advantageous in determining and optimizing the detection efficiency of vWCPCs. The Graetz model was
first used by Stolzenburg (1988) to examine the detection of ultrafine particles in CPCs. In recent years, COMSOL
Multiphysics® has been widely used to simulate coupled heat, mass, and momentum transfer problems associated
with complex geometries in CPCs. Moreover, COMSOL has advantages in interfacing with post-processing software
such as MATLAB$^{\text{TM}}$. A series of parametric analyses for butanol CPCs were simulated using COMSOL to investigate
the performance of particle activation and droplet growth (Hao et al., 2021; Attoui, 2018; Kangasluoma et al., 2015;
Barmpounis et al., 2018; Thomas et al., 2018). Our previous work (Hao et al., 2021) first demonstrated that the
COMSOL results neglecting the temperature dependence of vapor thermodynamic properties and axial diffusion,
agree with the Graetz solution used by Stolzenburg (1988). Considering temperature dependence of vapor
thermodynamic properties and axial diffusion can generate more accurate results that can guide the optimization of
CPC designs. Previous research using COMSOL to examine vWCPC performance has been limited. Bian et al. (2020)
compared two-stage and three-stage operating temperature methods for growth tubes and parameters such as flow rate
and temperature difference to obtain the ideal activation and final growth sizes. Mei et al. (2021) used COMSOL
aiding to examine how inlet pressure affects particle activation in the vWCPCs. However, the lack of thorough and
systematic examination of vWCPC performance using COMSOL leaves it unclear on how well the vWCPC will
perform in multiple complex research areas and applications, such as at reduced atmospheric pressure levels.

In this study, we thoroughly determined the saturation profile, activation efficiency, and droplet growth for varying
airborne operations through numerical simulation of laminar flow vWCPC in COMSOL and experimental validation
of a commercial vWCPC (TSI Model 3789). By mapping vWCPC performance in the modeling, the effects of various
operational factors, such as inlet pressures (0.3 - 1 atm), growth tube diameter and initiator length, and temperature
gradients on particle activation and droplet growth, were investigated. In addition, detailed modeling methods are
outlined below. The detection efficiency was also examined in the experimental and COMSOL modeling work. The
results of this study will guide further optimization of the performance of vWCPCs for accurate detection of particles
and atmospheric aerosol measurement applications.



## 2 Methods

### 2.1 Numerical simulation

#### 2.1.1 COMSOL setup

The finite element COMSOL simulation software (COMSOL Multiphysics 5.3a, COMSOL Inc, Stockholm, Sweden) can handle a variety of fields including, but not limited to, electromagnetics, fluid dynamics, heat transfer, chemical reactions, and structural mechanics. Here, a two-dimensional axisymmetric model is developed to simulate fluid flow in a cylindrical tube. The heat, momentum, and mass transfer equations are solved for incompressible parabolic flow. This COMSOL model follows the three-stage tube of the versatile water-based CPC (TSI Inc, Shoreview, MN, USA) described by Hering et al. (2017), which consists of a fully developed laminar flow tube that can be separated into a cool-wall conditioner region, a warm-wall initiator region, and a cool-wall moderator region, where r is the radial coordinate of the tube diameter and z is the axial coordinate of the tube length (**Fig. 1a**). At the inlet of the conditioner tube, sampled aerosols are fed and saturated with water vapor before entering the initiator region. The manufacturer provides two default cut-off diameter settings: 2 and 7 nm configurations based on the characteristics introduced by Kangasluoma et al. (2017). This study used the 7 nm configuration as an example to demonstrate the simulation mapping effort. The methodology can also be used for other targeted cut-off sizes. The tube diameter ($D$) is 6.3 mm, the conditioner length ($L_{con}$) is 73 mm, the initiator length ($L_{ini}$) is 30 mm, and the moderator length ($L_{mod}$) is 73 mm. The default settings are (**Table 1**): the conditioner temperature ($T_{con}$) is 30 °C, the initiator temperature ($T_{ini}$) is 59 °C, and the moderator temperature ($T_{mod}$) is 10 °C. The aerosol flow rate ($Q_v$) is 0.3 L min$^{-1}$. The relative humidity (RH) of inlet flow is set at 20%, and the water vapor is assumed to be saturated at the wall. The inlet pressure ($P$) is 1 atm. To investigate how the vWCPC performance depends on these parameters, we simulate a wide range of values for mapping the vWCPC geometry, working temperature conditions, and inlet pressure, as discussed in Section 2.1.4 for the tasks in this study.

In this COMSOL model, the coupled heat transfer and fluid flow are first solved by the conjugate heat transfer module, and then the mass transfer of the water vapor is solved based on the obtained temperature and flow field. Lastly, the particles are introduced from the inlet of the vWCPC and are considered diluted species that follow the convective diffusion equation in the simulation, which is numerically solved to calculate the temperature, supersaturation, and water vapor concentration profiles (**Fig. 1**). Regarding the initial and boundary value settings, the inflow temperature is set to an ambient temperature of 25 °C, and wall temperature is set to the default factory settings described above and changed as the experimental conditions changed for mapping performance. The simulated temperature and concentration fields are then imported into MATLAB$^{TM}$ R2022b (The MathWorks, Inc., Natick, MA, USA) and interpolated at any given point in the *r-z* plane. COMSOL Multiphysics 5.3a with MATLAB allows us to adjust geometry, modify physics settings, perform parametric studies, control solvers, and post-process the results.



### 2.1.2 Theory of particle activation


Particle activation is key to the evaluation of CPC performance. Particle activation within the vWCPC depends on the
degree of supersaturation, namely the saturation ratio ($S$) of water vapor, which is the ratio of the partial pressure of
the water vapor ($p$) to the saturation vapor pressure of the water vapor ($p_s$) for the given flow temperature ($T$),
calculated by

$$S = \frac{p}{p_s} \tag{1}$$

The spatial profile of $S$ within the vWCPC allows us to calculate the Kelvin effect, the homogeneous nucleation, and
further condensational particle growth, as discussed in Section 2.1.3. The Kelvin effect is dependent upon
thermodynamic principles and described as the Kelvin equivalent size ($D_{p,kel}$), the minimum diameter of a particle
that can be activated for condensation growth. It is determined by water surface tension ($\sigma$), water molecular volume
($v_m$), the Boltzmann constant ($k$), temperature ($T$) and the saturation ratio ($S$) calculated at each location within the
initiator,

$$D_{p,kel} = \frac{4\sigma v_m}{kT ln(S)} \tag{2}$$

The Kelvin equivalent size is inversely proportional to the distribution of saturation ratio, where the greater the
saturation ratio, the smaller the size of the particles that can be activated. In other words, the smaller the particle, the
higher the degree of supersaturation required to activate growth. When particle size ($D_p$) is above $D_{p,kel}$, the particle
can be successfully activated and grown by water vapor condensation, while when $D_p$ is below $D_{p,kel}$, the particles
cannot be activated. **Fig. 1b - 1e** show examples of saturation ratio, Kelvin equivalent size, water vapor concentration,
and temperature profiles within the simulated geometry at the default temperature condition of $T_{con}$ = 30 °C, $T_{ini}$ = 59
°C, $T_{mod}$ = 10 °C, respectively.

It is worth noting that $D_{p,kel}$ varies at different locations of the initiator due to the spatial variation of temperature,
surface tension and saturation ratio. We observed the potential particle activation in the moderator by simulation
results. However, only the Kelvin equivalent size in the initiator was considered in this work. Although the particles
were activated in the moderator, particle detection was unlikely due to droplet growth being the dominant water vapor
sink in the initiator. Thus, in reality, the actual supersaturation in the moderator may be different, and the activation
of smaller particles will be hindered by droplet growth (Hering et al., 2014; Hering et al., 2017). Note that the $D_{p,kel}$ in
the conditioner region is blank in color due to no particles being activated in this region ($S \leqslant 1$).      Particles near the
wall of the initiator, where there is a lower $S$, are more difficult to activate due to the larger $D_{p,kel}$. This difference can
be explained by the water mass and heat diffusivity differences. As the colder, water-saturated flow passes through
the growth tube, the mass transport of water vapor is faster than the heating of the flow from the wall because the mass
diffusivity of water vapor is higher than the thermal diffusivity of air, producing a maximum supersaturation of water
vapor at the centerline of the tube. As a result, the seed particles entering near the centerline of the growth tube are
activated in the warmer initiator. One comparison of the saturation ratio and Kelvin equivalent size along the centerline
($r$ = 0) is shown in **Fig. S1a and S1b.** We observed the appearance of a double-peaked saturation ratio curve. Again,





only the Kelvin equivalent size in the initiator was considered in this work due to insufficient water vapor and droplet
growth in the moderator. Note that our calculations do not include the solute effects, and we assume wettable insoluble
particles in the modeling.

The activation efficiency of particles with a size of $D_\mathrm{p}$ in vWCPC is derived using an approach similar to our previous
work (Hao et al., 2021), which is calculated by the ratio of the number concentration of the activated particles over
the total particle number concentration in the growth tube. The activation efficiency is calculated as

$$\eta_\mathrm{act} = \frac{\int_0^{R_\mathrm{act}} 2\pi r w N \, dr}{Q_\mathrm{v} N_0} \tag{3}$$

where $w$ is the velocity along the axial direction, $N$ is the concentration of particles, both at the axial location of z =
$Z_\mathrm{act}$. $R_\mathrm{act}$ is the maximum radius of the contour corresponding to $D_\mathrm{p,kel} = D_\mathrm{p}$. $N_0$ is the particle concentration at the
inlet of the conditioner. $Q_\mathrm{v}$ is the flow rate through the vWCPC. An example of activation efficiency as a function of
particle diameter $D_\mathrm{p}$ can be found in **Fig. S1c**. Note that the calculation of the activation efficiency in Eq. (3) does not
consider the diffusion loss of the particles in the conditioner. Since this model does not have sheath flow that
minimizes the diffusion losses and constrains the aerosols to the high supersaturation region, the activation efficiency
cannot reach 100%. On the activation curve, there are two points of interest: minimum activated size, $D_\mathrm{p,kel,0}$ (the
smallest size of particle that can be activated in the initiator), and 50% cut-off size, $D_\mathrm{p,kel,50}$ (the size of a particle with
50% activation efficiency extracted from the activation efficiency curve). $D_\mathrm{p,kel,50}$ is essential to the performance of
CPCs because it determines the general particle size range in which the CPC can confidently measure. $D_\mathrm{p,kel,50}$ can
be used as the representative of particle activation efficiency performance. Furthermore, note that negligible
homogeneous nucleation occurs in the growth tube of the initiator and moderator under all tested conditions in this
study, which means the total nucleation rate is equal to or less than one particle per second (1 s[-1]).

**2.1.3 Theory of droplet growth**
Once the particles are activated, their condensational growth along their trajectories in the initiator region was
simulated by numerically solving two coupled differential equations in MATLAB™. First, the evolution of droplet
diameter ($D_\mathrm{p}$) can be governed by (Seinfeld and Pandis, 2008; Wang et al., 2017)

$$\frac{dD_\mathrm{p}}{dt} = \frac{4 D_\mathrm{v}' M}{\rho} \frac{(C - C_\mathrm{d})}{D_\mathrm{p}} \tag{5}$$

where $M$ and $\rho$ represent the molecular weight and density of water, $D_\mathrm{v}'$ represents the modified diffusivity of the
water vapor accounting for the non-continuum effect of the particles and is given by $D_\mathrm{v}' = D_\mathrm{v} \left[ 1 + \frac{2 D_\mathrm{v}}{\alpha_c D_\mathrm{p}} \left( \frac{2\pi M}{RT} \right)^{1/2} \right]^{-1}$,
where $D_\mathrm{v}$ is the diffusivity of the water vapor, and $\alpha_c$ is the mass accommodation coefficient of water and is assumed
as 1. $C_d$ represents the equilibrium water concentration at the surface of the growth droplets and is given by $C_d =$





$C_s(T_d) exp(\frac{4\sigma M}{\rho R T_d D_p})$, where $C_s$ is saturation water concentration, $T_d$ is the droplet surface temperature, which is
governed by

$$\frac{dT_d}{dt} = \frac{3}{c_p \rho D_p}\left(H_{vap}\rho \frac{dD_p}{dt} - 4k_g' \frac{(T_d - T)}{D_p}\right) \tag{6}$$

where $c_p$, $\rho$, and $H_{vap}$ are the heat capacity, density, and heat of vaporization of water. $k_g'$ represents the modified
thermal conductivity of air accounting for the non-continuum effects in heat transfer and is calculated as $k_g' =$
$k_g\left[1 + \frac{2k_g}{\alpha_T D_p \rho_g c_{p,g}}\left(\frac{2\pi M_g}{RT}\right)^{1/2}\right]^{-1}$, where $M_g$, $\rho_g$, $c_{p,g}$, and $k_g$ are the molecular weight, density, heat capacity, and
thermal conductivity of air. $\alpha_T$ is the thermal accommodation coefficient and was assumed to be 1 (Seinfeld and
Pandis, 2008; Wang et al., 2017).

In the simulation of droplet growth, the Brownian motion of the particles inside the conditioner before activation is
neglected. Since there is also no electric field inside the conditioner, we assume that the particles move axially along
the vWCPC with a velocity of $w$. Therefore, Eqs. (5) and (6) can be converted to a function of axial location using
$w = dz/dt$, and the droplet size and droplet surface temperature at the end of the moderator ($z = L_{ini} + L_{mod}$) can be
calculated. To determine the final droplet growth size at the outlet of the moderator, the condensational growth of 8
nm particles as seed particles was studied along the centerline ($r = 0$) of the growth tube in this work. An example of
droplet growth size as a function of distance along the axis of the tube for the default temperature condition of $T_{con} =$
30 °C, $T_{ini}$ = 59 °C, $T_{mod}$ = 10 °C can be found in **Fig. S1d**. Note that in this simulation, we do not consider the
increase in the equilibrium vapor pressure due to warming of the flow from condensational heat release, which would
further reduce the droplet growth.

**227 2.1.4 Simulation plan**

**Table 1** summarizes the operating temperatures, inlet pressures, and geometric parameters for each simulation task
characterizing the vWCPC in this study. In Task 1, we first conducted two matrix combinations of absolute conditioner
and initiator operating temperatures, with an interval of 5 °C in each region, for a total of nine different combinations.
Task 2 investigates how raising or lowering the temperature midpoints ($T_{mid}$), the average value between the
conditioner temperature and initiator temperature ($\frac{T_{con}+T_{ini}}{2}$) affects particle activation and droplet growth. In
addition, different inlet pressures are also included when comparing different temperature midpoints. Task 3 examines
the effect of inlet pressure by comparing the default temperature of 30 °C and the conditioner temperature of 27 °C.
Tasks 4 and 5 further test how the vWCPC geometry, including tube diameter $D$, and initiator length $L_{ini}$, affects the
performance of the vWCPC. These simulations reveal optimal working conditions and effects for the influence of
each parameter.





**2.2 Experimental measurement**
The modified vWCPC 3789 (TSI Inc, Shoreview, MN, USA) was tested in this study. Operating flow, temperatures
and geometry can be found in Section 2.1.1 and our previous study (Mei et al., 2021). Two methods were used to
generate the test aerosol: an atomizer coupled with a furnace; and a glowing wire generator (GWG). Ammonium
sulfate (AS) has been commonly used for CPC characterization and was the tested material used in this study (Hering
et al., 2014; Kangasluoma et al., 2017). It was dissolved into deionized water for aerosol generation using atomization
techniques. To increase the aerosol number concentration for particles less than 30 nm, polydisperse AS aerosols were
also passed through a tube furnace generator (Lindberg/Blue, Thermal Scientific, TX, USA) to shift the size
distribution to a smaller size. A lab-built GWG was also used to generate aerosol particles in size range between 2.5
– 16 nm. More details about the generator can be found in Attoui (2022). Using the low-pressure testing setup shown
in **Fig. S2**, the counting efficiency of a vWCPC 3789 was measured between 0.5-0.9 atm for AS particles of 3 – 20
nm (mobility diameter) and NiCr oxidants of 2.5 -16 nm. The aerosol concentrations in this test were maintained in
the range of $2 \times 10^4 - 4 \times 10^4$ cm$^{-3}$. During the testing, the temperature variations in the conditioner and moderator were
less than 0.5 ℃, and the initiator temperature had a variation of 1 ℃. The y-axis error bar indicates the standard
deviation of the counting efficiency averaged over ∼ 5 min of sampling time at a 1 Hz sampling rate.

**3 Results and discussion**
**3.1 Comparisons of temperature-dependent particle activation and droplet growth performance**
Selection of appropriate operating temperatures in CPCs is essential because the supersaturation is significantly
temperature dependent, which affects particle activation and further droplet growth. In addition, the temperature
difference between different regions in CPCs is an important factor in controlling supersaturation. For this reason, the
minimum activation size for butanol-based CPCs is significantly impacted by the temperature difference between the
saturator and condenser and the raising or lowering of the temperature midpoints, as has been demonstrated by many
previous studies (Hermann and Wiedensohler, 2001; Kangasluoma and Attoui, 2019; Barmpounis et al., 2018; Kuang
et al., 2012). The results showed that in the butanol-based CPCs, the greater the temperature difference between the
saturator and condenser, the higher the degree of supersaturation, and the smaller particle could be activated.

The numerical COMSOL model was used to compare operating temperature-dependent particle activation and droplet
growth performance in the vWCPC, including minimum activated size ($D_{\mathrm{p,kel,0}}$), 50% cut-off size ($D_{\mathrm{p,kel,50}}$), and final
growth particle size at the outlet of the moderator along the centerline ($r = 0$) ($D_{\mathrm{d}}$), as shown in **Fig. 2**. Previous studies
confirmed that the centerline saturation rate is insensitive to the moderator wall temperature (Hering et al., 2014; Bian
et al., 2020). Thus, this study investigated moderator temperature ($T_{\mathrm{mod}}$) at the constant of 10 ℃, conditioner
temperature ($T_{\mathrm{con}}$) at the range of 25 – 35 ℃, initiator temperature ($T_{\mathrm{ini}}$) at the range of 55 – 65 ℃. Note that conditions
that can lead to a lower $D_{\mathrm{p,kel,50}}$ value and larger droplet growth size are favored for improving the performance of
the vWCPC.


Firstly, in order to compare the effect of the conditioner temperature $T_{con}$, we increased $T_{con}$ from 25 °C to 35 °C
while maintaining the same initiator temperature $T_{ini}$ and moderator temperature $T_{mod}$, $D_{p,kel,0}$ increased significantly
by 5.21, 3.32, and 2.27 nm at the initiator temperature $T_{ini}$ of 55, 60, and 65 °C, respectively, and $D_{p,kel,50}$ increased
significantly by 6.65, 4.16, and 2.75 nm at the initiator temperature $T_{ini}$ of 55, 60, and 65 °C, respectively. The final
droplet size $D_d$ decreased by approximately 1 μm at all the initiator temperature $T_{ini}$ of 55, 60, 65 °C. The lower
conditioner temperature provided higher saturation ratios in the initiator and more water vapor for particle growth,
which is also consistent with the previous growth tube simulation (Bian et al., 2020; Mei et al., 2021). Secondly, the
initiator temperature $T_{ini}$ was increased from 55 °C to 65 °C while maintaining the same conditioner temperature $T_{con}$
and moderator temperature $T_{mod}$, $D_{p,kel,0}$ was decreased by 1.90, 3.74, and 4.87 nm at the conditioner temperature
$T_{con}$ of 25, 30, and 35 °C, respectively, $D_{p,kel,50}$ was decreased significantly by 2.32, 3.74, and 6.21 nm at the
conditioner temperature $T_{con}$ of 25, 30, and 35 °C, respectively, and $D_d$ was increased by 2.9 μm at all the conditioner
temperature $T_{con}$ of 25, 30, and 35 °C.

By comparing all combinations, we can find that the activated size becomes smaller as the temperature difference
between $T_{con}$ and $T_{ini}$ increases, indicating that the temperature differences between the conditioner and initiator
dominate the particle activation. After comparing the temperature differences, we conclude that the higher the
temperature between these two regions, the better the particle activation. However, in the actual operation of the CPC,
one also needs to ensure that the self-nucleation in the growth tube is minimized ($<1$ s$^{-1}$) so that the CPC does not
report false particle counting. The homogeneous nucleation rate is less than $10^{-8}$ s$^{-1}$ at all tested conditions, meaning
that the temperatures can be further adjusted to optimize particle activation and droplet growth. Moreover, $D_d$ is the
greatest, with a maximum size of 12.20 μm, at the temperature setting of 25–65–10 °C among all these temperature
conditions. We also found that the effect of the initiator temperature on droplet growth was greater than that of the
conditioner temperature. Thus, the following section examines the effect of temperature midpoint on the vWCPC
performance.

**3.2 Effect of temperature midpoint on particle activation and droplet growth performance**
In addition to temperature difference, lowering the temperature midpoint was also found to cause higher
supersaturation. However, there is limited research on how the performance of the vWCPC changes under various
temperature midpoints and especially under different inlet pressures, which will be important for applications such as
atmospheric airborne deployment and environmental monitoring at elevated locations. Here, we compared the particle
activation and droplet growth performance for three different temperature midpoints (40 °C, 43 °C, and 46 °C) of
conditioner temperature (from 24 °C to 30 °C) and initiator temperature (56 °C to 62 °C) at a wide range of inlet
pressures from 0.3 atm to 1 atm, as shown in **Fig. 3**. The temperature difference $\Delta T$ between the conditioner and the
initiator was kept constant at 32 °C. The moderator temperature remained constant at 10 °C in all simulations.





Results show that the minimum activated size $D_{p,kel,0}$ decreases from 5.15 nm to 4.96 nm, and the 50% cut-off size
$D_{p,kel,50}$ decreases from 5.88 nm to 5.65 nm as the temperature midpoint decreases from 46 ℃ to 40 ℃, as shown in
**Fig. 3a and 3b**. Thus, a slight control of the minimum activation size can be achieved by lowering the temperature
midpoint. Higher supersaturation can explain this slight decrease in the initiator, which also agrees with the previous
growth tube WCPC simulation (Bian et al., 2020). On the other hand, a slight increase of 0.07, 0.1, and 0.14 nm occurs
in $D_{p,kel,0}$, and negligible change in $D_{p,kel,50}$ by reducing the inlet pressure from 1 to 0.3 atm under three temperature
midpoints of 40 ℃, 43 ℃, and 46 ℃. This slight increase is due to a low peak supersaturation caused by the decrease
in inlet pressure. Since water vapor transport is faster than heat transport, the decrease in pressure affects the location
of the peak supersaturation, whereas the degree of the supersaturation does not change significantly.

In **Fig. 3c**, we show that the droplet growth is not significantly dependent on raising or lowering the temperature
midpoint. By lowering the temperature midpoint by 6 ℃, $D_d$ becomes smaller by approximately 14%. When studying
the effect of inlet pressure on the $D_d$, unlike $D_{p,kel,0}$ and $D_{p,kel,50}$, $D_d$ decreases substantially from 1 to 0.3 atm, by
approximately 45%. Limited by the optical chamber design of the commercial vWCPC, the droplets smaller than 8
μm may not gain sufficient pulse signal to get counted. Thus, when operating under lower inlet pressure, the apparent
cut-off size of vWCPC may increase and needs to be further determined. The reduced pressure strongly affects the
final droplet growth size, likely due to the faster water vapor and heat transport at reduced pressure. The thermal and
mass diffusivity is inversely proportional to the pressure in the growth tube, resulting in insufficient time for droplet
growth. In addition, we found that with the lower inlet pressure, the final droplet size reduced more notably. For
example, $D_d$ decreased from 10.6 to 10.4 μm (by 0.2 μm) as pressure reduced from 1.0 to 0.9 atm, while $D_d$ decreased
from 7.2 to 6.1 μm (by 1.1 μm) as pressure reduced from 0.4 to 0.3 atm. The difference can be explained by the
competition between heat and water vapor transport. The mass transport of water vapor is faster than the heating flow
from the wall because the mass diffusivity of water vapor is higher than the thermal diffusivity of air. Therefore, by
reducing the inlet pressure, water vapor transport becomes even faster than heat transfer due to the water vapor
diffusivity and air thermal diffusivity being inversely proportional to the pressure, further shortening the time for
particle growth at high supersaturation. This observation demonstrates for the first time how the final droplet size is
affected by raising or lowering temperature midpoints at standard and various reduced inlet pressure conditions in the
vWCPC.

**3.3 Effect of inlet operation pressure on particle activation and droplet growth performance**
With the advantages of safe, eco-friendly and readily available distilled water as working fluid in the vWCPC,
applying the vWCPC in various inlet pressures will expand broader applications such as atmospheric airborne aerosol
measurements. Here, we examined the effect of inlet pressure on minimum activated size, $D_{p,kel,0}$, 50% cut-off size,
$D_{p,kel,50}$, and final growth particle size at the outlet of the moderator along the centerline ($r = 0$), $D_d$ from 0.3 to 1 atm
for two different temperature settings: the conditioner, initiator, and moderator temperatures were 30, 59, and 10 ℃
and 27, 59, and 10 ℃ in **Fig. 4**.




**Figs. 4a** and **4b** show $D_{p,kel,0}$ and $D_{p,kel,50}$ as a function of inlet pressure, relatively greater (2 - 3%) $D_{p,kel,0}$ was
observed at reduced inlet pressures at both conditioner temperatures of 27 °C and 30 °C. This increase is because the
supersaturation value at reduced pressure is lower than the saturation profile under standard conditions. We also found
that the saturation profile peaked earlier, closer to the entrance of the initiator in the low-pressure condition. In
addition, greater $D_{p,kel,50}$ is observed at reduced inlet pressures due to the reduction of the saturation peak at both
conditioner temperatures of 27 °C and 30 °C. Again, the difference at reduced inlet pressure can be explained by the
competition from heat transfer and water vapor transport, as discussed in Section 3.2. For this reason, greater $D_{p,kel,50}$
was observed at reduced inlet pressures. This reduction of saturation peaks is also associated with the growing droplet
size decreasing with the decrease in the operating pressure. Again, lowering the conditioner temperature while
maintaining the same temperature difference between the initiator and the moderator provided higher saturation ratios
in the initiator over all pressure ranges.

**Fig. 4c** shows the final droplet size as a function of inlet pressure. When the conditioner temperature is 27 or 30 °C, a
lower final droplet size (∼ 40 % reduction in the droplet size) was observed at a reduced inlet pressure of 0.3 atm,
indicating insufficient droplet growth happens at low-pressure conditions, which is consistent with the previous study
that insufficient droplet growth becomes more significant under low-pressure operation (Mei et al., 2021).

Furthermore, in addition to showing consistent results with the previous study (Mei et al., 2021), our simulations
enhance guidance for aircraft applications under extreme conditions, which can be achieved by simulating low
atmospheric pressure at 0.3 atm. As shown in Section 3.5, by comparing with experimental results, our simulations
can provide more accurate estimates of particle activation and droplet growth to guide vWCPC for low-pressure
applications.

**3.4 Effect of tube diameter and initiator length on particle activation and droplet growth performance**
The geometry in CPCs also impacts the CPC activation performance and particle growth due to the changed
supersaturation and temperature profile in the tube, as discussed in previous studies (Hao et al., 2021; Hering et al.,
2014). Here, we examined how the tube diameter $D$ and the length of initiator $L_{ini}$ in the vWCPC may affect the
minimum activated size, $D_{p,kel,0}$, 50% cut-off size, $D_{p,kel,50}$, and final growth particle size at the outlet of the moderator
along the centerline ($r = 0$), $D_d$ under default temperature $T_{con}$–$T_{ini}$–$T_{mod}$ of 30–59–10 °C, standard inlet pressure
and reduced pressure of 0.5 atm using the numerical COMSOL model. Again, one needs to note that conditions that
can lead to a lower $D_{p,kel,0}$ and $D_{p,kel,50}$ value and larger droplet growth size are favored for improving the performance
of the vWCPC.

We examined four values of $D$ from 4 to 8 mm, and five values of $L_{ini}$ from 10 to 50 mm, shown in **Fig. 5** and **Fig. 6,**
respectively. The results indicate that a smaller $D$ can slightly decrease in $D_{p,kel,0}$ approximately 0.03 nm, while no



noticeable changes on $D_{p,kel,50}$ at the standard pressure (**Figs. 5a and 5b**). By reducing the tube diameter, the flow
speed in the tube increases under the same flow rate, reducing the residence time of the condensed water vapor. This
reduction in residence time suppresses homogeneous nucleation in the initiator. Unlike our previous study on CPCs
(Hao et al., 2021), the homogeneous nucleation rate is minimal in vWCPC and has no impact on the temperature
difference compared to butanol-based CPCs. For this reason, this suppressed homogeneous nucleation has limited
effects on $D_{p,kel,0}$ and $D_{p,kel,50}$. However, the increase of the flow speed will significantly limit the time for droplet
growth, as will be discussed later. At the reduced pressure of 0.5 atm, a smaller $D$ can slightly decrease in
$D_{p,kel,0}$ approximately 0.08 nm, and a slight decrease of approximately 0.03 nm on $D_{p,kel,50}$ (**Figs. 5a and 5b**). Overall,
the reduction in pressure plays a more critical role in negatively impacting CPC performance for relatively large tube
diameters. Note that buoyancy effects (Roberts and Nenes, 2005) may be critical for large temperature differences if
the tube diameter is too large, which is not discussed in the study.

On the other hand, we found that reducing $L_{ini}$ leads to limited effects on $D_{p,kel,0}$ and $D_{p,kel,50}$, except for the shortest
initiator length of 10 mm at the standard pressure (**Figs. 6a and 6b**). The effect of these relatively long initiator lengths
is limited because the degree of supersaturation is determined by the absolute temperature of the tube flow. The
temperature difference did not change in the standard pressure and reduced pressure, leaving both $D_{p,kel,0}$ and $D_{p,kel,50}$
unchanged. However, at the initiator length of 10 mm, $D_{p,kel,0}$ and $D_{p,kel,50}$ increase significantly due to insufficient
water vapor diffusion before passing through the next moderator region, resulting in a lower peak supersaturation
along the centerline than for longer initiators operating at the same temperature (Hering et al., 2014). At reduced
pressure, $D_{p,kel,0}$ and $D_{p,kel,50}$ have no noticeable changes at all tested initiator lengths, however, this is due to the
sufficient diffusion of water vapor, from which the water transport is faster than at the standard pressure. Again, if the
initiator is longer, the difference in peak supersaturation will be negligible, while the peak temperature along the
centerline and the amount of added water vapor will be higher. Thus, for relatively short initiators, such as 20 mm
used in the simulation, one can provide all the necessary water vapor to create the same peak supersaturation as for
the longer initiators. However, the droplet growth size will be smaller (**Fig. 6c**), mainly due to the shorter growth time
discussed later.

With regard to the performance of particle growth, an increased $D$ and an increased $L_{ini}$ are beneficial for improving
the performance of particle growth in vWCPC at both standard and reduced pressure (**Figs. 5c and 6c**). An increased
$D$ implies a decrease in the flow velocity through the high saturation region, greatly increasing the time for particle
growth and contributing to the sufficient growth of the particles. **Fig. 5c** shows that the final droplet sizes increase
significantly from 6.72 μm to 13.88 μm when $D$ is increased from 4 mm to 8 mm at the standard pressure and increase
from 4.92 to 10.78 μm at the reduced pressure of 0.5 atm. The final droplet size is found to be 2 – 3 μm smaller than
the standard pressure at the reduced pressure. Similarly, a longer $L_{ini}$ also leads to a larger droplet growth size. The
final droplet size increases from 8.66 μm to 11.26 μm when $L_{ini}$ is increased from 10 mm to 50 mm at the standard
pressure and from 7.40 to 8.29 μm at the reduced pressure of 0.5 atm (**Fig. 6c**). This increase is likely due to the longer
growth time of the longer initiator. Also, we found that the final droplet size increases much faster at shorter initiator





lengths than at lengths above 20 mm, which tells us that the droplet size is more susceptible to the effects of initiator
length below 20 mm. This difference also means that having a longer length does not further enhance the final size of
the particle growth.

In addition to the performance of particle activation, it is crucial to evaluate the droplet growth performance of
complex geometries in the vWCPC. The time that allows the activated particle to grow in the initiator and moderator
is an important droplet growth kinetics assumption, representing the vWCPC performance of droplet growth. We use
$t_g$ to represent allowed particle growth time, approximated with Eq. (7).

$$t_g \sim D^2 L^* / Q_v \tag{7}$$

where $L^*$ indicates the length of the initiator and moderator beyond the point of activation. This equation can explain
that the residence time is impacted more by the change in tube diameter than the initiator length. The allowed particle
growth time as a function of final growth particle size at the outlet of the moderator along the centerline ($r = 0$), $D_d$ is
shown in **Fig. S3**. The longer the allowed particle growth time, the larger the droplet growth size. Based on this droplet
growth time shown, the vWCPC geometry of $D$ and $L_{ini}$ are not independent variables if we consider the droplet
growth for further particle detection.

**3.5 Experimental measurement validation of detection efficiency**
Experimental validation is essential for simulation work in terms of the accuracy of the simulation model and the
correctness of the underlying trends. Furthermore, validation and good agreement will provide well-guided approaches
for future applications. Therefore, we compare the experimental and simulation results of the counting efficiency and
detection efficiency of vWCPC for two configurations of 2 nm and 7 nm at different conditioner and initiator
temperature settings and different low-pressure conditions in **Fig. 7**.

As the experimental results in a previous study (Mei et al., 2021) are shown in **Fig. 7a,** the counting efficiency of
vWCPC 3789 varied with different working pressures (500, 700 and 910 hPa) when the conditioner temperature is 27
°C, the initiator temperature is 59 °C, and the moderator temperature is 10 °C. The results indicate that the counting
efficiency slightly decreases with the decrease in the operating pressure of 500, 700 and 900 hPa, which shows the
same trend in **Fig. 7b.** In addition, the cut-off size in both experimental and simulation results are in the range of 5 -
7 nm, which is also an acceptable range within error when compared to commercial vWCPC detection efficiency.

**Fig. 7c** and **7d** compare the counting efficiency and detection efficiency versus particle size from experimental and
simulation results under initiator temperatures of 75 and 90°C and pressure of 910 and 500 hPa for the 2 nm
configuration. As expected, the detection efficiency of both experimental and simulated results is lower at the
temperature $T_{con} - T_{ini} - T_{mod}$ of 7-75-10 °C at a lower pressure (at 500 hPa). When the temperature $T_{con} - T_{ini} - T_{mod}$
is 7-90-10 °C, the higher detection efficiency is seen, and the effect of inlet pressure becomes insignificant. However,
it is not feasible to maintain 90 °C when operating under lower pressure, such as 500 hPa. Thus, the default 2 nm



setting in vWCPC can only be operated near sea level. One should note that although we do not present many
simulations for the 2 nm configuration, what we learned from the modeling results with the 7 nm setting will guide
future simulations with the 2 nm setting.

By comparing with counting efficiency curves, the present simulations can more realistically represent the $D_{\mathrm{p,kel}}$ for
7 nm vWCPC, which also achieved a good agreement with the 2 nm setting. Thus, from the merits of the results of
this work, we can find that this work not only provides guidance for 7 nm, but this trend can also help guide one for
other desired cut-off sizes.

### 463   4 Conclusions

This study evaluated the particle activation and droplet growth performance of a commercial versatile water CPC
using COMSOL in combination with MATLAB data processing. In addition, validation experiments on the detection
efficiency of the commercially modified vWCPC (TSI 3789) agreed with the simulation work. Increasing the
temperature difference between $T_{\mathrm{con}}$ and $T_{\mathrm{ini}}$ and lowering the temperature midpoint can enhance particle activation
at both standard and reduced ambient pressure conditions. However, the lack of droplet growth becomes more
significant under low-pressure operations, which might affect the apparent counting efficiency of the vWCPC due to
the limited measurable size range of the optical chamber. Additionally, reducing the diameter of the growth tube
slightly improved particle activation without enhancing the droplet growth, while increasing the initiator length had a
limited effect on improving the performance of the vWCPC at both standard and reduced pressure.

This simulation realistically represents the $D_{\mathrm{p,kel}}$ for 7 nm vWCPC and shows that the current growth tube geometry
is an optimized choice for aerosol measurements. This study will guide further vWCPC performance optimization for
applications requiring precise particle detection and atmospheric aerosol monitoring. Furthermore, the developed
simulation capability provides a vital tool for the aerosol community to understand the effects of temperature and
pressure on vWCPC behavior. The knowledge gained will guide the field deployment of vWCPC on the ground level
and airborne measurements. Thus, several future experimental studies will be carried out to investigate the
performance of the vWCPC.

**Data availability**
The vWCPC data in the study are available upon request to Fan Mei (fan.mei@pnnl.gov).

**Author contributions**
WH, FM, and YW designed the research. FM carried out the measurements. WH led the simulation and data analyses.
WH led the writing, with significant input from FM and YW as well as further input from all other authors. SH, SS,
BS, and JT provided suggestions on the revision.



**Competing interests**
Susanne Hering has a commercial interest in the success of the vWCPC instrument.

**Acknowledgments**
Hao and Wang are partially supported by NSF award 2132655.

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



**Table of Nomenclature**
$c$: Molecular concentration of the water vapor [mol m$^{-3}$]
$C_d$: Equilibrium water concentration at the surface of the growth droplets [mol m$^{-3}$]
$C_s$: Saturation water concentration [mol m$^{-3}$]
$c_p$: Heat capacity of the water [J K$^{-1}$ kg$^{-1}$]
$c_{p,g}$: Heat capacity of air [J K$^{-1}$ kg$^{-1}$]
$D_v$: Diffusivity of the water vapor [m$^2$ s$^{-1}$]
$D_v'$: Modified diffusivity of the water vapor [m$^2$ s$^{-1}$]
$D$: Diameter of the growth tube in vWCPC [m]
$D_p$: Particle size [m]
$D_d$: Final growth droplet size in vWCPC [m]
$D_{p,kel}$: Size of particle that can be activated according to the Kelvin equation [m]
$D_{p,kel,0}$: Smallest size of particle that can be activated in the vWCPC [m]
$D_{p,kel,50}$: Size of particle that has a 50% activation efficiency [m]
$H_{vap}$: Heat of vaporization of water [J kg$^{-1}$]
$k$: Boltzmann constant, $1.38 \times 10^{-23}$ [J K$^{-1}$]
$k_g$: Thermal conductivity of air [W m$^{-1}$ K$^{-1}$]
$k_g'$: Modified thermal conductivity of air [W m$^{-1}$ K$^{-1}$]
$L_{con}$: Length of the conditioner [m]
$L_{ini}$: Length of the initiator [m]
$L_{mod}$: Length of the moderator [m]
$L^*$: Length of the initiator and moderator beyond the point of activation [m]
$m$: Molecular mass of water [kg]
$M$: Molecular weight of water [kg mol$^{-1}$]
$M_g$: Molecular weight of air [kg mol$^{-1}$]
$n$: Molecular concentration of the water vapor [molecules m$^{-3}$]
$N$: Concentration of the particles at the axial location of $z = Z_{act}$ [particles m$^{-3}$]
$N_0$: Concentration of particles at the inlet of the conditioner [particles m$^{-3}$]
$p$: Partial pressure of the water vapor [Pa]
$p_s$: Saturation vapor pressure of the water vapor [Pa]
$P$: Inlet pressure in vWCPC [Pa]
$Q_v$: Flow rate through the vWCPC [m$^3$ s$^{-1}$]
$r$: Radial coordinate of the tube diameter of the vWCPC [m or as otherwise explicitly designated]
$R$: Gas constant [J mol$^{-1}$ K$^{-1}$]
$R_{act}$: Maximum radius of the contour corresponding to $D_{p,kel} = D_p$ [m]
$S$: Saturation ratio [1]





$t$: Allowed particle growth time [s]
$T$: Flow temperature in the CPC [K]
$T_\mathrm{con}$: Conditioner temperature [K]
$T_\mathrm{ini}$: Initiator temperature [K]
$T_\mathrm{mid}$: Temperature midpoint corresponding to $\frac{T_\mathrm{con}+T_\mathrm{ini}}{2}$ [K]
$T_\mathrm{mod}$: Moderator temperature [K]
$T_\mathrm{d}$: Droplet surface temperature [K]
$v_\mathrm{m}$: Molecular volume of the water vapor [m$^3$]
$w$: Velocity along the axial direction in the vWCPC [m s$^{-1}$]
$z$: Axial coordinate of the tube length of the vWCPC [m or as otherwise explicitly designated]
$Z_\mathrm{act}$: Axial location corresponding to $r = R_\mathrm{act}$ [m]
$\alpha_\mathrm{c}$: Mass accommodation coefficient of water [1]
$\alpha_\mathrm{T}$: Thermal accommodation coefficient of air [1]
$\eta_\mathrm{act}$: Activation efficiency [1]
$\rho$: Density of water [kg m$^{-3}$]
$\rho_\mathrm{g}$: Density of air [kg m$^{-3}$]
$\sigma$: Surface tension of water [N m$^{-1}$]
$\Delta T$: Temperature difference between conditioner temperature and initiator temperature [K]





**Table 1. Parameters of vWCPC for different simulation tasks. Note that the default settings of VWCPC are: the conditioner**
**temperature ($T_{con}$) is 30 ℃, the initiator temperature ($T_{ini}$) is 59 ℃, and the moderator temperature ($T_{mod}$) is 10 ℃. The**
**aerosol flow rate ($Q_v$) is 0.3 L min$^{-1}$. The relative humidity (RH) of inlet flow is set at 20%, and the water vapor is assumed**
**to be saturated at the wall. The inlet pressure ($P$) is 1 atm.**

| Task | $T_{con}$ (°C) - $T_{ini}$ (°C) | $T_{mod}$ (°C) | $T_{mid}$ (°C) | $P$ (atm) | $D$ (mm) | $L_{ini}$ (mm) |
|---|---|---|---|---|---|---|
| 1 | (25, 30, 35) - (55, 60, 65) | 10 | - | 1 | 6.3 | 30 |
| 2 | 24 - 56, 27 - 59, 30 - 62 | 10 | 40, 43, 46 | 0.3 - 1 | 6.3 | 30 |
| 3 | 27 - 59, 30 - 59 | 10 | - | 0.3 - 1 | 6.3 | 30 |
| 4 | 30 - 59 | 10 | - | 0.5, 1 | 4, 5, 6.3, 8 | 30 |
| 5 | 30 - 59 | 10 | - | 0.5, 1 | 6.3 | 10, 20, 30, 40, 50 |


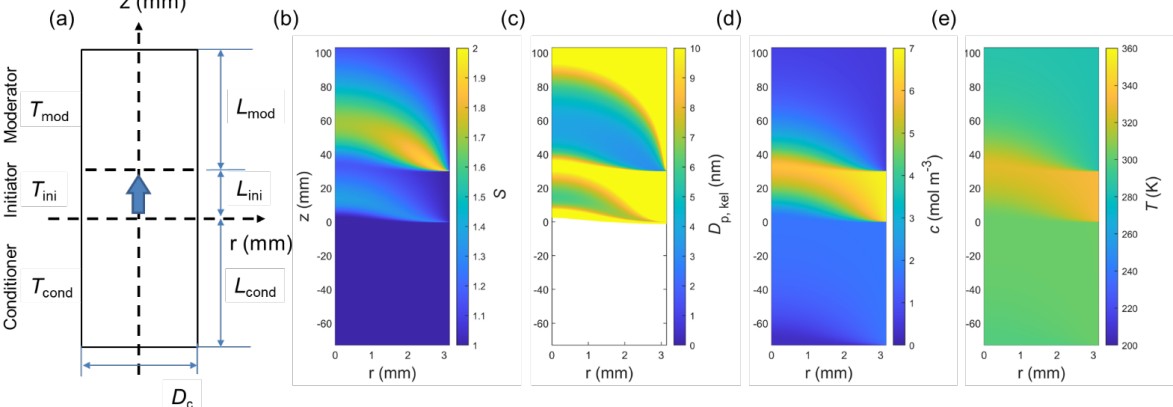



**Figure 1.** Geometry of vWCPC used in COMSOL simulation and spatial distribution of saturation ratio and Kelvin equivalent size under 30−59−10 °C temperature setting. (a) Geometry of the vWCPC used in COMSOL simulation, (b) Spatial distribution of saturation ratio ($S$, color contour plot), (c) Spatial distribution of Kelvin equivalent size ($D_{\mathrm{p,kel}}$, color contour plot), (d) Spatial distribution of water vapor concentration ($c$, color contour plot), and (e) Spatial distribution of temperature ($T$, color contour plot). Note that the color of $D_{\mathrm{p,kel}}$ in the conditioner region is blank because no particles are activated in this region.





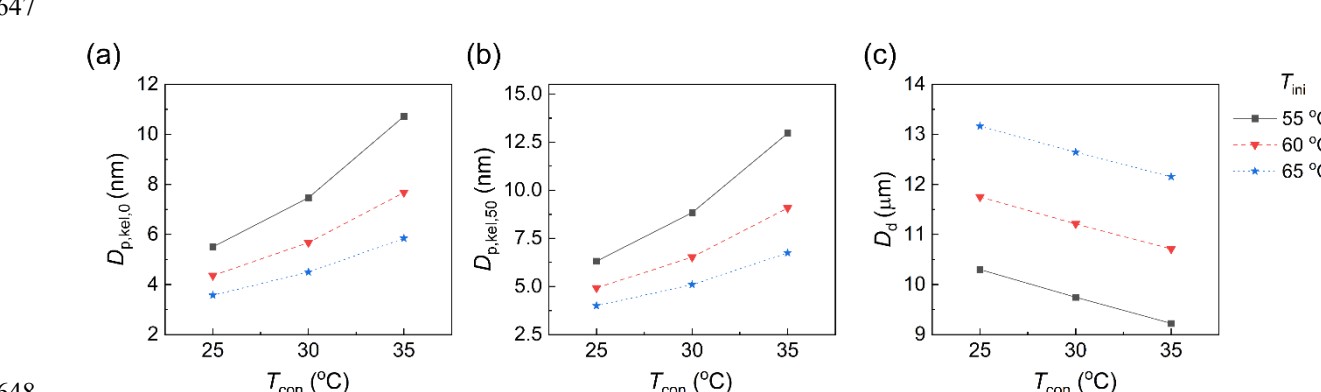


**Figure 2. Effect of conditioner ($T_{con}$) and initiator temperature ($T_{ini}$) on (a) minimum activated size, $D_{p,kel,0}$, (b) 50%**
**cut-off size, $D_{p,kel,50}$, and (c) final growth particle size at the outlet of the moderator along the centerline ($r = 0$), $D_{d}$.**
**The condensational growth of 15 nm particles was tested as seed particles.**

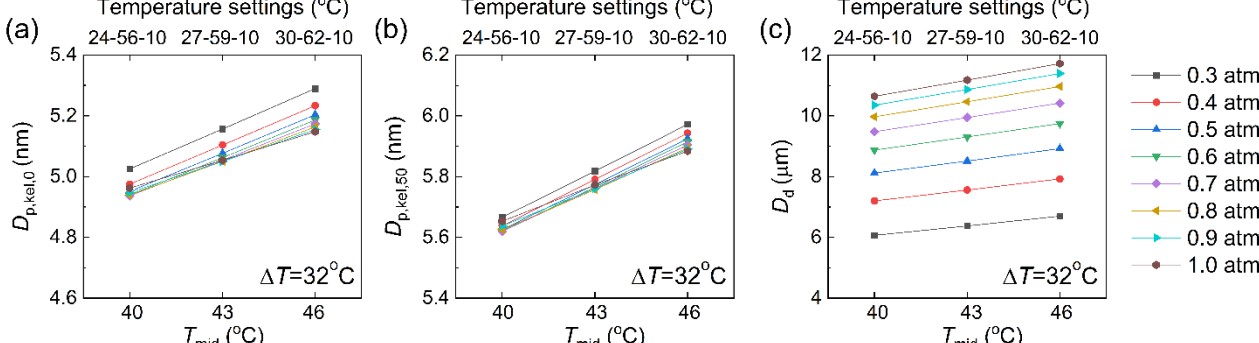


**Figure 3. Effect of temperature midpoints at 40 °C, 43 °C, and 46 °C at $T_{con}$-$T_{ini}$-$T_{mod}$ of 24−56−10 °C, 27−59−10 °C and 30−62−10 °C with a constant temperature difference of 32 °C on (a) minimum activated size, $D_{p,kel,0}$, (b) 50% cut-off size, $D_{p,kel,50}$, and (c) final growth particle size at the outlet of the moderator along the centerline ($r = 0$), $D_d$. The condensational growth of 8 nm particles was tested as seed particles.**

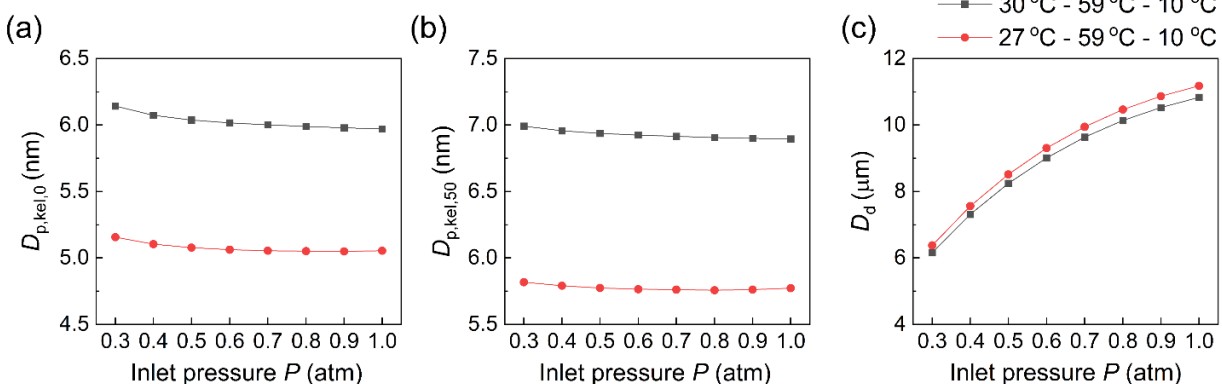

**Figure 4. Effect of inlet operation pressure at $T_{con}$-$T_{ini}$-$T_{mod}$ of 27–59–10 °C and 30–59–10 °C on (a) minimum activated size, $D_{p,kel,0}$, (b) 50% cut-off size, $D_{p,kel,50}$, and (c) final growth particle size at the outlet of the moderator along the centerline ($r = 0$), $D_d$. The condensational growth of 8 nm particles was tested as seed particles.**



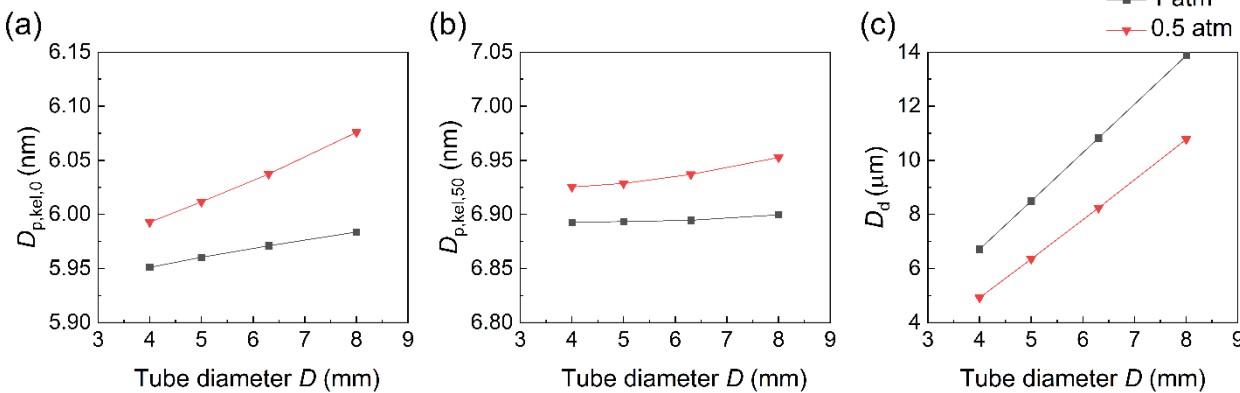

**Figure 5. Effect of tube diameter ($D$) at 0.5 atm and 1 atm on (a) minimum activated size, $D_{p,kel,0}$, (b) 50% cut-off size, $D_{p,kel,50}$, and (c) final growth particle size at the outlet of the moderator along the centerline ($r = 0$), $D_d$. The condensational growth of 8 nm particles was tested as seed particles.**



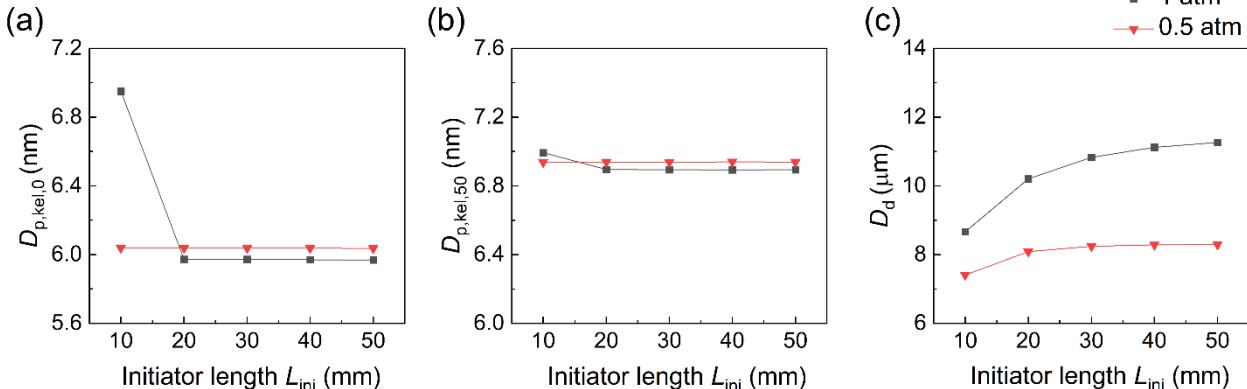

**Figure 6. Effect of initiator length ($L_{ini}$) at 0.5 atm and 1 atm on (a) minimum activated size, $D_{p,kel,0}$, (b) 50% cut-off size, $D_{p,kel,50}$, and (c) final growth particle size at the outlet of the moderator along the centerline ($r$ = 0), $D_d$. The condensational growth of 8 nm particles was tested as seed particles.**



**Figure 7. vWCPC operation validation: (a) the counting efficiency of experimental results as a function of particle size under the conditioner temperature of 27 ℃ and pressure of 910, 700, and 500 hPa for the 7 nm configuration, (b) the detection efficiency of simulation results as a function of particle size under the conditioner temperature of 27 ℃ and pressure of 0.9, 0.7, and 0.5 atm for the 7 nm configuration, (c) the detection efficiency of experimental results as a function of particle size under initiator temperatures of 75 and 90℃ and pressure of 910 and 500 hPa for the 2 nm configuration, and (d) the detection efficiency of simulation results as a function of particle size under initiator temperatures of 75 and 90℃ and pressure of 0.9 and 0.5 atm for the 2 nm configuration.**