# Peer review of "Mapping the performance of a versatile water-based"

_Atmospheric Measurement Techniques, 2023_

## Author Comment (AC1)

**RESPONSE TO COMMENTS**
**Title: Mapping the performance of a versatile water-based condensation particle counter (vWCPC) with COMSOL simulation and experimental study**
**Journal: Atmospheric Measurement Techniques**
**Ref: AMT-2023-45**

*Reviewer #1:*

*Comment: The only thing which is missing in this interesting work is the response time vers the diameters of the tube.*

**Response:** Thank you for your positive feedback. In this study, we varied the diameter of the growth tube to examine how it affects the particle activation and growth under the assumption that the introduction of aerosols is under steady state. Therefore, the response time of the vWCPC is not considered, and it does not affect the particle activation efficiency. However, if the aerosols are introduced into the vWCPC in a pulsed manner, the wider diameter of the growth tube will lead to a longer residence time of the particles, which will eventually lead to a longer response time. Assuming that the flow rate through the growth tube is constant, the residence time of particles in the growth tube is proportional to the volume, and therefore, to the cross section area of the growth tube or the square of tube diameter. Under the same assumption, we would expect that the response time is also proportional to the square of the tube diameter.

*Comment: Few questions and comments on the other hand:*

*The details given by the authors about the changes made in the 'modified commercial water CPC' are not clear or not enough. Indeed the authors are talking about changes in a 'commercial' version of the CPC. That could be taken to mean by the reader as some thing doable by any body who has a TSI 3789 CPC. Few details about the 'modified commercial water CPC' will be very helpful for the reader. What was the change made in the CPC compared to the commercial version? What are the benefits or advantages provided by these changes?It will help the readers to have an exhaustive description of the modified commercial instrument.*

**Response:** We have comprehensively revised and enhanced the details of the modified commercial water CPC in Section 2.2. This incorporates more thorough information from the previously published work by Mei et al., 2021. The main benefit of these revisions is to make this version of the water CPC usable for low-pressure applications, which is the goal of this study to characterize the performance of the vWCPC at low pressures. To clarify the name, we use "modified vWCPC" throughout the paper. A detailed description of the modified water CPC is provided in Section 2.2, as below.

"The modified vWCPC 3789 (TSI Inc, Shoreview, MN, USA) was tested in this study. Given that the standard commercially available vWCPC 3789 is not specifically designed for low-pressure applications, some modifications were made to the instrument for this study. First, the vWCPC 3789 was tested to ensure its vacuum tightness, and the exhaust line was filtered and returned back to the make-up flow line after a flow buffer. Second, we added pressure transducers to the inlet and exhaust lines of the vWCPC 3789 to monitor the inline pressure variation. Note that the aerosol flow rate through the condenser tube and optical particle detector

was 0.3 L min$^{-1}$. When we operated with 0.6 L min$^{-1}$ inlet aerosol flow, we blocked the make-up flow port. Details of operating flow, temperatures and geometry are provided in Section 2.1.1. Further specifics can be found in our previous study (Mei et al., 2021)."

*Why this particular temperature of the initiator 59°C  (default value) rather than 60°C for example?*

**Response:** Although our experiments did not include specific tests at initiator temperatures of 59 °C and 60 °C, our simulations showed that particle activation and droplet growth performance are similar within this temperature range. Consequently, we adhered to the default temperature of 59 °C for the TSI vWCPC throughout this study. To further investigate the potential impacts of temperature differences on particle activation and droplet growth performance, we evaluated and conducted tests under a diverse set of temperature conditions.

*Comment: Is their a large difference if one takes 60°C. What is the precision on the temperature measurement?*

**Response:** Our simulation work did not show significant differences in the vWCPC performance between 59 °C and 60 °C. However, it is important to note that the simulated temperature may not exactly reflect the actual temperature in the vWCPC. As for temperature measurement accuracy, the sensor in the vWCPC may present an error margin of ±1 °C, and it may potentially degrade over time. In this study, we adhered to the default temperature precision settings in the simulation and measurements.

*Comment: The authors should cite the previous  work of Ahn & Liu*

*Ahn, Kang-Ho and Liu, B. Y. H. (1990) Particle activation and droplet growth processes in condensation nucleus counter--I. Theoretical background J. Aerosol Sci. 21, 249-261.*

*Ahn, Kang-Ho and Liu, B. Y. H. (1990): Particle activation and droplet growth processes in condensation nucleus counter—II. Experimental study. J. Aerosol Sci. 21, 263-275.*

**Response:** Thank you for providing the references. We have cited these two valuable papers in our revised manuscript.

We thank the reviewer for his comments. This has improved our manuscript and we look forward to the paper being accepted for publication.

---

## Author Comment (AC2)

**RESPONSE TO COMMENTS**
**Title: Mapping the performance of a versatile water-based condensation particle counter (vWCPC) with COMSOL simulation and experimental study**
**Journal: Atmospheric Measurement Techniques**
**Ref: AMT-2023-45**
*Reviewer #2: [General Comments]*

*Comment:The manuscript describes simulation results for modified water-based condensation particle counters operated with different instrumental parameters. The study is useful to guide the development of particle instruments targeting non-standard applications such as airborne deployments and investigation of very small nucleation mode particles. The study is overall presented well although the manuscript would benefit from careful language editing. Some sentences are very long and lack clarity. The part discussing the experimental verification is very short and could be extended.*

**Response:** Thank you for the positive feedback. We have revised the language to make it clearer and more concise. The section on experimental validation has been expanded and incorporated more fully into our discussion. The responses to the comments are as follows.

*Comment:Given the discussion of very small differences in activation diameters and other quantities - is there any way to assess the uncertainty of the derived quantities from numerical algorithms within COMSOL? Are there any aspects in the model where assumptions might break down or are questionable? Even though the derived values show physically reasonable trends a discussion of practical relevance of those small variations should be included.*

**Response:** Thank you for raising this point. COMSOL, like any other numerical simulation software, is based on numerical algorithms that inevitably introduce some level of uncertainty. Uncertainty may arise due to discretization errors, rounding errors, modeling errors, or due to the simplification and idealization of the real-world problem to be analyzed. Therefore, it is important to validate the simulation results against experimental data to confirm their accuracy and reliability.

In this work, our COMSOL simulation takes into account heat, fluid, and momentum. Limitations of this modeling approach likely lie in the computational capacity and the difference from actual flow and temperature profile distribution in the growth tube (due to non-ideal temperature control and system configuration). However, given the agreement between the simulation and experimental results, the simulation is trustworthy and can offer valuable guidance on particle activation and droplet growth within the vWCPC. Once established the COMSOL program and integrated it with MATLAB, the data produced are reproducible and accurate. Future improvements could involve considering more complex scenarios (e.g., the non-ideal heat transfer and flow condition) by modifying initial and boundary conditions.

We added the limitations to the end of the Conclusion section:

"However, limitations of this modeling approach likely lie in the computational capacity and the difference from actual flow and temperature profile distribution in the growth tube (due to non-ideal temperature control and system configuration)."

*Comment:l. 181: How significant is the limitation of this assumption? It might be good to discuss at least qualitatively how the results might be affected by partially soluble particles - especially since the experiments are done with AS.*

**Response:** Thank you for pointing this out. We have qualitatively incorporated the new Figure S2 into the discussion of solute effects. The results show that the effect of solutes on the final droplet size is negligible when taken into account, and has little effect (off by 1%) during the droplet growth simulation. In this case, the final droplet size does not change due to soluble particles. Therefore, we clarify this sentence as:

"Note that the solute effect is negligible under the water-based condensation particle growth and is not included in the following simulation. As shown in **Fig. S2**, around 1% variation was observed in the final droplet size by adding the solute effect into the droplet growth simulation."

*Comment:l. 203: "can be governed by..." sounds very vague - please state the conditions where this formula is valid.*
**Response:** Thank you for your asking. This formula is valid under some assumptions, such as steady state, isolated droplet, no solute effect and curvature effects, etc. We changed this sentence to:
"the evolution of droplet diameter ($D_p$) can be estimated by…"

*Comment:l. 234: This sentence is unclear, please rephrase.*
**Response:** The sentence has been modified to make it clearer. The new changes are:
"Task 3 examines the effect of inlet pressure at the default conditioner temperature of 30 ℃ and the conditioner temperature of 27 ℃."

*Comment:l. 246: What temperature and flow rate was the furnace operated at?*
**Response:** The furnace temperature is 500 ℃. The furnace flow rate is 1.5 lpm. We changed this sentence to:
"To increase the aerosol number concentration for particles less than 30 nm, polydisperse AS aerosols were also passed through a tube furnace generator at the temperature of 500 ℃ and flow rate of 1.5 lpm (Lindberg/Blue, Thermal Scientific, TX, USA) to shift the size distribution to a smaller size."

*Comment:l. 249: Figure S2 could be incorporated into the main paper since the setup is central to the experiment.Does the validation hold towards lower pressures? For tropospheric airborne research pressures down to ~200hPa might be relevant.*
**Response:** Thank you for your suggestion. We have included Figure S2 in the main manuscript as the new Figure 2. The validation results, as demonstrated in Figure 7, show a slight decrease in counting efficiency with reducing operating pressure at 51, 71, and 91 kPa. This trend agrees with our simulation findings. Our experimental system can be operated above 500 hPa (51 kPa) only. Thus, the lower pressures will be investigated in our future experimental measurements.

*Comment:l. 252/253. What figure does this sentence refer to?*
**Response:** Thank you for pointing this out. The figure refers to Figure 8. We have deleted this sentence from this section and moved it to Section 3.5.

*Comment:l. 275ff: this sentence is long and confusing.*
**Response:** Thank you for your suggestion. We have split this sentence into two sentences.
"Firstly, in order to compare the effect of the conditioner temperature $T_{con}$, we increased $T_{con}$ from 25 ℃ to 35 ℃ while maintaining the same initiator temperature $T_{ini}$ and moderator

temperature $T_{mod}$. The results show that $D_{p,kel,0}$ increased significantly by 5.21, 3.32, and 2.27 nm at the initiator temperature $T_{ini}$ of 55, 60, and 65 °C, respectively, and $D_{p,kel,50}$ increased significantly by 6.65, 4.16, and 2.75 nm at the initiator temperature $T_{ini}$ of 55, 60, and 65 °C, respectively."

*Comment:l. 331: heating flow --> heat flow*
**Response:** We have changed "heating flow" to "heat flow".

*Comment:l.339ff: The correlations with inlet pressure are already shown in the previous subsection - I would recommend limiting the discussion in 3.2 to the T-effects and discussing all pressure-related results in 3.3. Alternatively, merge and overall shorten the two subsections.*
**Response:** Thank you for your suggestion. However, we prefer to maintain the discussion of pressure within Section 3.2 to uphold the clarity and structure of both Sections 3.2 and 3.3. As seen in Figures 3 and 4, Figure 3 primarily discusses the effects of temperature, with a supplemental presentation of pressure effects to aid reader comprehension. This incorporation of pressure effects is valuable in understanding the broader goals of our study. On the other hand, Figure 4 is designed to help readers concentrate on the effects of pressure specifically.

*Comment:l. 381: Is this change in D in any way of practical relevance? See comment above.*
**Response:** Thank you for your asking. In our experimental setup, we were limited by the vWCPC modification capabilities and hence, unable to adjust the tube diameter and initiator length. However, this exploration of parameters presents a direction for our future studies regarding the optimization of the vWCPC geometry for activating and growing smaller particles. In the current research, we have provided simulation results that will help guide these future endeavors and strategic planning.

*Comment:Fig 6 a/b: if the y-axis would be scaled as in Fig 5a/b would there be dependence on initiator length for L>20mm visible?*

**Response:** Should we scale the y-axis as we did in Fig. 5a/b (Now Fig. 6a/b), we would observe extremely minute differences, even smaller than those shown in Fig. 5 (Now Fig. 6). Therefore, we consider this trend to be negligible at this point in Fig. 6a/b (Now Fig. 7a/b). Our focus lies predominantly on the droplet growth aspect, as the key message of this study is the strong dependence of droplet growth on the tube geometry.

*Comment:l.429ff.: is it obvious what causes plateau-like shape of the curves shown in Fig. S3? What happens between 0.7 and 1s? Is there a better phrase for "Allowed particle growth time"?*
**Response:** The plateau-like shape of the curves might be due to the more significant impact of the tube diameter compared to the initiator length. The plateau is where we simulated various initiator lengths. This 0.7 - 1s interval represents stages with the same tube diameter but different initiator lengths. Overall, this plateau-like shape demonstrates that the tube diameter has a more substantial influence on droplet growth performance than the initiator length. More importantly, Fig. S3 indicates that the longer the allowed particle growth time, the larger the droplet growth size. The vWCPC geometry of $D$ and $L_{ini}$ are not independent variables if we consider the droplet growth for further particle detection.

The concept of "allowed particle growth time" has been carried over from our previous CPC paper (Hao et al., 2021). We aim to maintain consistency between the findings of these two studies.

*Comment:l.438 the phrasing "...for two configurations of 2nm and 7nm..." is unclear - maybe better use "..vWCPC set in the 2nm and the 7nm-configuration..." or something along those lines.*

**Response:** Thank you for your suggestion. We changed this sentence to:

"Therefore, we compare the experimental and simulation results of the counting efficiency and detection efficiency of vWCPC set in two default configurations (2 nm and 7 nm) at different conditioner and initiator temperature settings and different low-pressure conditions in **Fig. 8**."

*Comment:l. 441/Fig 7.: The reference to Mei et al 2021 as source of Fig 7a should also be given in the figure caption. The 30C TSI curve in Fig 7a either requires an explanation or should be left out if not relevant for this study.*

**Response:** We have incorporated the referenced source into the figure caption for accurate attribution. Furthermore, we removed the 30 ℃ TSI curve from this figure to maintain clarity and focus on the primary data.

*Comment:Sec. 3.5: In this section a mix of pressure units atm and hPa is used, this should straightened out for consistency, ideally the authors should resort to SI units throughout the paper unless there is a good reason not to.*

**Response:** To ensure consistency, we have standardized the units throughout the manuscript to kPa. Using kPa is particularly advantageous for atmospheric measurements and monitoring. Alongside the revisions in Section 3.5, we have also updated Figures 1 through 8 for consistency, ensuring that all components of our paper align with this standardized unit.

*Comment:The comparison of model and measurement results should really be shown in a single plot for each of the CPC settings. The different axis scales for the different panels make it difficult to see the key message.*

**Response:** Thank you for your suggestion. However, we have opted for the current presentation due to the following reasons: First, there are different intervals between the experimental and simulation data. The experimental data is obtained at intervals of a few nanometers, whereas the simulation data is collected at smaller intervals. Hence, presenting both data sets in a single plot could result in an overly cluttered chart, making it difficult to distinguish and interpret the data. Second, this work primarily relies on simulations, with one example provided to show the agreement in trends between the simulation and experimental data. This approach keeps the focus on the simulation work, while still demonstrating its practical relevance and validity. However, in response to your feedback, we have revised Figure 8. We have changed the units and efficiency scale to improve clarity and removed any redundancies. This revision should enhance the readability.

*Comment:l. 466: Please clarify, what does "commercially modified vWCPC" mean? Is this instrument commercially available in this form or has it been modified from its standard configuration for the purpose of those experiments?*

**Response:** The commercially modified vWCPC is a vWCPC that has been modified from its standard configuration for the purposes of the experiments in this study. The main benefit of these modifications is to make this version of the water CPC usable for low-pressure applications, which is the goal of this study to characterize the performance of the vWCPC at low pressures. To clarify the name, we use "modified vWCPC" throughout the paper. We also have comprehensively revised and enhanced the details of the modified water CPC in Section 2.2. This incorporates more thorough information from the previously published work by Mei et al., 2021. See below.

"The modified vWCPC 3789 (TSI Inc, Shoreview, MN, USA) was tested in this study. Given that the standard commercially available vWCPC 3789 is not specifically designed for low-pressure applications, some modifications were made to the instrument for this study. First, the vWCPC 3789 was tested to ensure its vacuum tightness, and the exhaust line was filtered and returned back to the make-up flow line after a flow buffer. Second, we added pressure transducers to the inlet and exhaust lines of the vWCPC 3789 to monitor the inline pressure variation. Note that the aerosol flow rate through the condenser tube and optical particle detector was 0.3 L min$^{-1}$. When we operated with 0.6 L min$^{-1}$ inlet aerosol flow, we blocked the make-up flow port. Details of operating flow, temperatures and geometry are provided in Section 2.1.1. Further specifics can be found in our previous study (Mei et al., 2021)."

*Comment:l. 471 It should be discussed in how far this improvement is practically relevant.*
**Response:** Indeed, our experimental setup is limited by the vWCPC modification capability to adjust the tube diameter and initiator length. However, in the current study, our simulation results indicate that the current growth tube ($D$ = 6.3 mm and $L_{ini}$ = 30 mm) is an optimized choice for the current vWCPC flow and temperature settings. These simulation results will provide important guidance and direction for our future practical work. Some modifications are made in the Conclusion Section:

"Furthermore, the developed simulation capability provides a vital tool for the aerosol community to understand the effects of temperature, pressure, and geometry on vWCPC behavior."

*Comment:I don't quite understand the statement "... without enhancing the droplet growth...". Doesn't Fig 5 show a reduction in growth for smaller tube diameters?*
**Response:** We have revised this sentence to avoid misunderstanding.

"Additionally, reducing the diameter of the growth tube slightly improved particle activation but significantly reduced the droplet growth, while increasing the initiator length had a limited effect on improving the performance of the vWCPC at both standard and reduced pressure."

*Comment:References should be listed with DOI numbers to ensure proper linking in online documents*
**Response:** Thank you for your suggestion. We have added all DOI numbers to the references.

We thank the reviewer for his / her comments. This has improved our manuscript and we look forward to the paper being accepted for publication.

---

## Author Response (AR2)

Dear Chuck,

Thank you very much for your suggestions. We have revised Fig 8. We also updated the reference format and removed the figure title page in the combined manuscript file. Please take a look and let us know if you have further suggestions.

Best regards,
Fan